# Contribution of Dopamine Transporter Gene Methylation Status to Cannabis Dependency

**DOI:** 10.3390/brainsci10060400

**Published:** 2020-06-23

**Authors:** Anna Grzywacz, Wojciech Barczak, Jolanta Chmielowiec, Krzysztof Chmielowiec, Aleksandra Suchanecka, Grzegorz Trybek, Jolanta Masiak, Paweł Jagielski, Katarzyna Grocholewicz, Blazej Rubiś

**Affiliations:** 1Independent Laboratory of Health Promotion, the Pomeranian Medical University in Szczecin, 11 Chlapowskiego St., 70-204 Szczecin, Poland; o.suchanecka@gmail.com; 2Department of Head and Neck Surgery, Poznan University of Medical Sciences, Fredry 10, 61-701 Poznan, Poland; barczak.woj@gmail.com; 3Department of Hygiene and Epidemiology, Collegium Medicum, University of Zielona Góra, Zyty 28 St., 65-046 Zielona Gora, Poland; chmiele1@o2.pl (J.C.); chmiele@vp.pl (K.C.); 4Department of Oral Surgery, Pomeranian Medical University in Szczecin, 72 Powstańców Wlkp. St., 70-111 Szczecin, Poland; g.trybek@gmail.com; 5Neurophysiological Independent Unit, Department of Psychiatry, Medical University of Lublin, 20-093 Lublin, Poland; jolantamasiak@wp.pl; 6Department of Nutrition and Drug Research, Faculty of Health Science, Jagiellonian University Medical College, Grzegorzecka 20, 31-501 Krakow, Poland; pawel.jagielski@gmail.com; 7Department of Interdisciplinary Dentistry, Pomeranian Medical University, 72 Powstańców Wlkp. St., 70-111 Szczecin, Poland; katarzyna.grocholewicz@pum.edu.pl; 8Department of Clinical Chemistry and Molecular Diagnostics, Poznan University of Medical Sciences, 49 Przybyszewskiego St., 60-355 Poznan, Poland; blazejr@ump.edu.pl

**Keywords:** dopamine transporter gene, *DAT1*, cannabis, epigenetics, dependency, CpG sites

## Abstract

The susceptibility to cannabis dependency results from the influence of numerous factors such as social, genetic, as well as epigenetic factors. Many studies have attempted to discover a molecular basis for this disease. However, our study aimed at evaluating the connection between altered methylation of the dopamine transporter gene (*DAT1*) promoter CpG sites and cannabis dependency. In the cases of some DNA sequences, including the *DAT1* gene region, their methylation status in blood cells may reflect a systemic modulation in the whole organism. Consequently, we isolated the DNA from the peripheral blood cells from a group of 201 cannabis-dependent patients and 285 controls who were healthy volunteers and who were matched for age and sex. The DNA was subjected to bisulfite conversion and sequencing. Our analysis revealed no statistical differences in the general methylation status of the *DAT1* gene promoter CpG island between the patients and controls. Yet, the analysis of individual CpG sites where methylation occurred indicated significant differences. These sites are known to be bound by transcription factors (e.g., SP1, p53, PAX5, or GR), which, apart from other functions, were shown to play a role in the development of the nervous system. Therefore, *DAT1* gene promoter methylation studies may provide important insight into the mechanism of cannabis dependency.

## 1. Introduction

Epigenetics refers to the processes that modify the regulation of genes without disrupting the DNA sequence. These processes may result in distinguishing a phenotype [1,2,3,4,5]. Consequently, genes might be turned on or off, which is also associated with exposure to certain agents, such as psychoactive substances. DNA methylation has been associated with behavioral abnormalities [6,7]. Accordingly, epigenetic alterations, including methylation status, might reflect environmental conditions and a person’s lifestyle. Therefore, epigenetic alterations may be used as biological markers that demonstrate metabolic dysfunctions [4,8,9]. Altogether, epigenetics may become a vital aspect of clinical diagnostics, as it encompasses the regulation of gene expression mediated by DNA methylation, nucleosome structure and positioning, posttranslational modifications of nucleosome histones, histone replacement, and RNA interference [10]. We understand that epigenetic alterations function to repress or activate genes, but since they are not fully understood, they may have other functions.

One of the most commonly studied and well-known epigenetic mechanisms is DNA methylation. It involves the addition of a methyl group to position C5 of cytosine at the CpG site [3,7] by DNA methyltransferase (DNMT) to form 5-methylcytosine (5-mC). 5-mC is an epigenetic marker that allows the expression of some genes but represses others [11], and may be passed on to the succeeding generations by means of cell division. Methylation alters chromatin packaging and makes it more condensed and consequently less available for transcription factors in general.

The regulation of DNA methylation has been analyzed in numerous physiological and behavioral phenotypes in animal models. Notably, it has been proven to be involved in neuronal and brain development [12,13,14]. In human studies, methylation dysregulation has been observed in individuals with substance dependency, anxiety, depression, autism, schizophrenia, and bipolar disorder [6,14,15,16].

Substance dependency involves long-term behavioral abnormalities that occur in response to repeated exposure to psychoactive substances in susceptible individuals. It is a multifactorial syndrome involving a complex interplay between genes and the environment. The data suggest that the underlying mechanisms regulating these persistent behavioral abnormalities involve changes in gene expression throughout the brain’s reward circuitry, particularly in the mesolimbic dopamine system. In the past decade, investigations have revealed the genes potentially involved in the risk of substance dependency using genome-wide association studies. These investigations have also established the crucial role of epigenetic mechanisms in mediating the enduring effects of drug abuse on the brain in animal models of dependency. Genetic variants have been proven to be involved in the onset and development of dependence on psychoactive substances. Interactions between environmental and genetic risk factors have also been proven to contribute to the onset and development of dependency, as environmental risk factors tend to have a greater effect in children with a genetic vulnerability. Thus, given the link between genetics and one’s environment, epigenetics may play a crucial role in dependency pathogenesis [17].

Marijuana, which has high levels of tetrahydrocannabinol (THC), is one of the most popular psychoactive substances. Globally, over 180,000,000 people use cannabis, and approximately 9% of them become dependent. When individuals begin using cannabis in adolescence, their dependency on it increases up to 16%, and reaches 50% if cannabis is used daily [18,19]. The studies on the impact of THC on epigenetic modifications showed altered histone methylation and acetylation in mouse immune cells that changed the expression of genes involved in the immune response [20]. The study of striatal mRNA levels in adolescent and adult mouse offspring with parental THC exposure showed a downregulation of gene promoters and changes in mRNA expression connected to differentially methylated region (DMR)-associated genes participating in glutamatergic synaptic regulation within the nucleus accumbens [21]. The impact of the mother’s and the infant’s exposure to THC during pregnancy was analyzed in the context of methylation changes in the dopamine receptor 4 *DRD4* gene. The study found no evidence of epigenetic modifications. A higher rate of DNA methylation was observed when the methylation status of other genes from the dopaminergic system was analyzed in a group of adult cannabis users in the dopamine receptor 2 (*DRD2*) and the neural cell adhesion molecule (*NCAM*) [22].

As mentioned above, the dopaminergic system plays a central role in the development of substance dependency. The concentration of extracellular dopamine levels is driven by its transporter encoded by the dopamine transporter (*DAT1*) gene [23]. We are unaware of any studies concerning the methylation status of the dopamine transporter gene in cannabis-dependent subjects with the exception of studies regarding that issue in the context of alcohol dependency [24,25,26], striatal dopamine transporter availability in ADHD (attention-deficit hyperactivity disorder) [27], gambling [28], and impulsivity [29] in animal studies.

Due to the crucial role the dopaminergic system plays in all substance use disorders, we looked to analyze the methylation status of 33 CpG sites located in the dopamine transporter gene promoter region in a group of cannabis dependent subjects.

Modulation of DNA methylation, both at specific GpC sites or at sites in the genome as a whole, is supposedly associated with numerous health outcomes. When cancer patients are examined, most of these changes are found at the tissue level in a tissue-specific manner. However, there are still some concerns regarding methylation changes in white blood cells (WBCs), as they may be a useful biomarker for various diseases. Epidemiological studies showed possible associations between global leucocyte DNA methylation and some cancers or schizophrenia [30,31]. It may be that those alterations observed in blood cells simply reflect some risk factors, such as demographics (age, gender, race) or environmental exposures (synthetic or organic air pollutants), and other risk factors (drugs, physical activity and diet, body size, and cigarette smoke or alcohol consumption). Therefore, large prospective studies are necessary to evaluate methylation status as a risk factor or as a result of substance dependency.

## 2. Materials and Methods

### 2.1. Samples

The study group consisted of 201 male patients diagnosed with a cannabis dependency according to the International Classification of Diseases, 10th Revision (ICD-10) (mean age = 27.28, SD = 6.06). The dependent subjects were recruited from the substance use disorder and dependency unit at the Residential Inpatient Treatment Center in the province of Lubuskie, Poland, after abstaining from drugs for a minimum of three months. None of the subjects were receiving pharmacotherapy.

The control group comprised of 285 healthy male volunteers matched for age (mean age = 21.91, SD = 4.08). The patients and controls were all Caucasians from the same region of Poland. The study was carried out at the Independent Laboratory of Health Promotion, Pomeranian Medical University in Szczecin. The protocol of the study was accepted by the Bioethics Committee (KB-0012/106/16) and informed written consent was provided by all participating individuals. All of the participants were also examined by a psychiatrist.

### 2.2. Methylation Status Assessment

DNA was extracted from peripheral blood leukocytes using a DNA isolation kit (A&A Biotechnology, Gdynia, Poland) as previously described [24] and stored at −20 °C. A bisulfite modification of 250 ng DNA was performed using the EZ DNA Methylation Kit (Zymo Research, Orange, CA, USA) per the manufacturer’s instructions. A methylation-specific PCR assay was carried out in a Mastercycler epgradient S (Eppendorf, Hamburg, Germany).

Primer oligonucleotides were obtained from Genomed.pl (Warsaw, Poland). Primer sequences were designed using MethPrimer version 1.0 software (Chinese Academy of Medical Sciences, Beijing, China, https://www.urogene.org/cgi-bin/methprimer/methprimer.cgi). The status of *DAT1* promoter (ENSG00000142319) was assessed by a PCR using primers specific to a fragment of the gene, i.e., DATF: 5′-GGTTTTTGTTTTTTTTATTGTTGAG-3′; DATR: 5′-AAATCCCCTAAACCTAATCCC-3′. To successfully amplify the 447 bp fragment covering 33 CpG sites in the *DAT1* gene promoter, the PCR conditions were as follows: initial denaturation (94 °C/5 min), followed by 35 cycles (94 °C/61 °C/72 °C, 25 s each step), and a final elongation at 72 °C for 5 min. The concentration of magnesium chloride ions was 2.5 mM. After amplification assay, the PCR products were subjected to previously described sequencing [24]. Briefly, samples were both verified by sequencing using the BigDye v3.1 kit (Applied Biosystems, Darmstadt, Germany) and separated by an ethanol extraction using ABI Prism 3130XL (Applied Biosystems, Darmstadt, Germany) in a 36 cm capillary in a POP7 polymer, using the reverse primer.

Sequencing chromatograms were analyzed using 4Peaks version 1.8 software (Mek & Tosj, Amsterdam, the Netherlands, https://nucleobytes.com/4peaks/index.html). The methylation of cytosine was considered positive when the G/A + G ratio reached at least 20% of the total signal (Figure 1).

### 2.3. Assessment of the Ability to Bind Transcription Factors

For the analysis of potential transcription binding sites, the *DAT1* promoter region sequence was loaded as a query for PROMO verion 3.0 software (Universitat Politècnica de Catalunya, Barcelona, Spain, http://alggen.lsi.upc.es/cgi-bin/promo_v3/promo/promoinit.cgi?dirDB=TF_8.3). The analysis was conducted using only sites and human transcription factors. Notably, when using PROMO, weight matrices are constructed from the known binding sites extracted from version 8.3 of the TRANSFAC database (http://genexplain.com/transfac/#section0) to identify potential binding sites in sequences. The ability of transcription factors to bind individual regions was assessed with a different binding accuracy, i.e., 100%, 95%, or 85%, which corresponds to the rate of similarity between putative and consensus sequences for a given transcription factor.

### 2.4. Statistical Analysis

The data were analyzed using the chi-squared test, with *p* < 0.05 considered as statistically significant (GraphPad Prism version 8.0 for Windows, GraphPad Software, San Diego, CA, USA, www.graphpad.com). The Bonferroni correction for multiple testing was applied to obtain the Bonferroni critical value.

## 3. Results

We evaluated the effect of cannabis dependency on the *DAT1* promoter methylation status in patients’ peripheral blood leucocytes in a side-by-side comparison with healthy individuals. We noticed that both groups presented a similar general methylation level of the CpG island. After we further explored the status of individual CpG sites, we observed significant methylation status differences in positions shown in Table 1. Four of these positions were methylated at a higher level (positions 1, 6, and 28), while nine showed an inverse trend (positions 3, 5, 13, 19, 22, and 25). Deeper analysis using the Bonferroni correction for multiple tests showed that the difference in CpG site methylation was greater, especially in positions 1 and 6.

Since high methylation of the gene promoter usually has a negative impact on the binding affinity of transcription factors (TFs) [32], we analyzed the TFs’ ability to bind individual regions with different similarity rates, i.e., 100%, 95%, or 85%. As demonstrated, all the sites with significantly different methylation statuses in both groups showed a capacity to bind different TFs that play multiple functions in cell metabolism (Table 2). Notably, one of the factors that might interact with the target sequence in a methylation-status-dependent manner was PAX5, as it was recognized for its role in neural development [33].

## 4. Discussion

The legalization of the use of cannabis for medical and recreational purposes has encouraged regular usage to become increasingly popular. It is one of the most frequently used psychoactive substances worldwide [19], and as a result, the number of research studies covering all aspects of the dependency on the substance, including epigenetic modifications, has grown. In the present study, we analyzed 33 CpG sites located in the promoter region of the *DAT1* gene in cannabis-dependent subjects and in controls. The results of our analyses showed significant differences between both groups—methylation changes are not similar across all sites. In comparison with the control group, some sites in the dependent subjects were hypermethylated, while some were hypomethylated. Further, the assessment of the ability of transcription factors to bind indicated sites revealed a substantial number of those regulators of gene expression.

Three of the analyzed sites were found to be possible PAX5 transcription factor binding sites (positions 3, 22, and 33), and all of them were hypomethylated in the group of dependent subjects. This is crucial, since PAX5 is a transcription factor associated with numerous processes including the development of the nervous system [34]. Unfortunately, we did not analyze the expression of the *DAT1* gene in our study group, so we are unable to fully anticipate what role this finding may have in relation to dopaminergic transmission. Additionally, the transcription’s binding ability revealed that the CpG island covers a sequence that may be bound by ligand-bound GR (position 6). As we demonstrated, this was the site that showed a significant difference in methylation status after the Bonferroni correction. This may be of vital importance since glucocorticoids mediate nervous system functions in substance dependency [35]. In response to this, we concluded that hindering glucocorticoid responses could change drug-induced reactions, including the modulation of drug-related learning and memory. Additionally, we postulated that GR may become a promising target in substance use disorders and dependency therapy [36,37].

As of yet, no studies regarding the *DAT1* methylation status have been performed in the context of cannabinoid dependency, and it is difficult to say if any direct association may be revealed. Further studies on animal models with the use of *DAT1* knock-out animals or chromatin immunoprecipitation (ChIP) experiments should be performed to reveal which TF or other proteins may bind DNA in the *DAT1* promoter region. However, since neural development and functionality is determined throughout one’s lifetime, it could not produce expected results. With this said, among possible TFs, which may bind the *DAT1* promoter in selected sites, SP1 can be found. In general, SP1 is a type of low specificity transcription factor with a wide spectrum of target sequences.

There is the possibility, however, that the *DAT1* methylation status might either predispose or reflect exposure to substances in dependent individuals. Therefore, we do not know if it is more of a prognostic or diagnostic marker. However, we do know that, functionally, epigenetic alterations affect the expression of genes, as measured by RNA and protein synthesis; this may in turn affect the cellular structure and function and consequently affect the whole body’s metabolism and behavior. With regard to drug-taking and -seeking behavior, the preclinical literature demonstrated that changes in the expression of genes, such as brain-derived neurotrophic factor (*BDNF*) [38] and *OPRM1* (the μ-opioid receptor gene) [39,40], alter the reinforcement properties of drugs including alcohol, cocaine, and heroin. Thus, epigenetic changes and resultant changes in gene expression may also contribute to our mechanistic understanding of substance dependencies.

Peripheral tissues have important limitations with respect to generalizability to other tissues of interest, such as the brain. However, there is a growing amount of evidence that proves that the numerous epigenetic changes found in peripheral leukocytes and transformed lymphoblasts also correspond to alterations in the brain cells [41]. Most importantly, a study performed in the previous year demonstrated that peripheral *DAT1* promoter methylation may be a predictive factor of dopamine transporter availability in the striatum [27].

Another important factor that should be considered when analyzing the epigenetic effects of THC exposure is the age of a given exposure. Recent animal studies [42,43] showed that the hippocampus, the amygdala, and the nucleus accumbens are impacted differently by chronic THC consumption, depending on age. Histone modifications in particular regions were also affected differently. The adolescent brain showed changes leading to transcriptional repression, whereas the effect on adults showed transcriptional activation. In adolescent hippocampus and nucleus accumbens, the initial repression was followed by a counterbalancing response. The authors suggested that this may have resulted from the adolescent brain being more vulnerable to the adverse effects of THC [42]. Similarly, adolescent exposure was shown to alter plasticity genes, which are important in the development of cognitive deficits. The process was mediated by histone modifications and the altered prefrontal cortex [43].

Conclusively, we think that *DAT1* methylation as well as methylation in general should be more extensively investigated in the context of substance dependency pathogenesis. Future studies should involve women and not only men, as in our case, as well as subjects of non-Caucasian origin. Because genetic and epigenetic markers vary by sex and race, the conclusions that can be drawn from this study are limited. Similarly, expression studies of *DAT1* should be performed to fully evaluate the impact of epigenetic modifications and other genes, as well as their role in dependency. There have been some reports showing that blood derived samples reveal a significantly lower expression of *DAT1* in substance-dependent patients than in controls [44], but without significant alterations in *DAT1* methylation [44,45]. However, both studies included opioid-dependent patients, which may constitute a crucial difference to one’s cannabinoid dependency. Findings for *DAT1* in other substance use disorders suggested broader implications for this subset of psychiatric disorder. This is crucial given the high rates of comorbidity among psychiatric disorders and the potential for comorbidities as confounding factors [46,47]. Therefore, it seems that DAT1 gene promoter methylation studies may provide important insights into the mechanism of the development of substance dependency.

There is currently very little research concerning epigenetic, and specifically, methylation changes in addiction. To the best of our knowledge, there is no study analyzing the methylation of the few genes in the same group of subjects that also incorporates a full phenotypic description of addiction (i.e., substance(s), age of onset, number of relapse, and many others), psychometric testing, and single nucleotide polymorphism (SNP) data, so the question of replicability is still open.

## 5. Conclusions

The results of our analyses of 33 CpG sites located in the promoter region of the *DAT1* showed significant differences between both analyzed groups (cannabis dependent and controls)—methylation changes are not similar across all sites. In comparison with the control group, some sites in the dependent subjects were hypermethylated, while some were hypomethylated. Further, the assessment of the ability of transcription factors to bind indicated sites revealed a substantial number of those regulators of gene expression. Three of the analyzed sites were found to be possible PAX5 transcription factor binding sites (positions 3, 22, and 33), and all of them were hypomethylated in the group of dependent subjects. This is crucial, since PAX5 is a transcription factor associated with numerous processes including the development of the nervous system. Additionally, the transcription’s binding ability revealed that the CpG island covers a sequence that may be bound by ligand-bound GR (position 6). As we demonstrated, this was the site that showed a significant difference in methylation status after the Bonferroni correction. Therefore, we conclude that *DAT1* gene promoter methylation studies may provide important insight into the mechanism of cannabis dependency.

## Figures and Tables

**Figure 1 brainsci-10-00400-f001:**
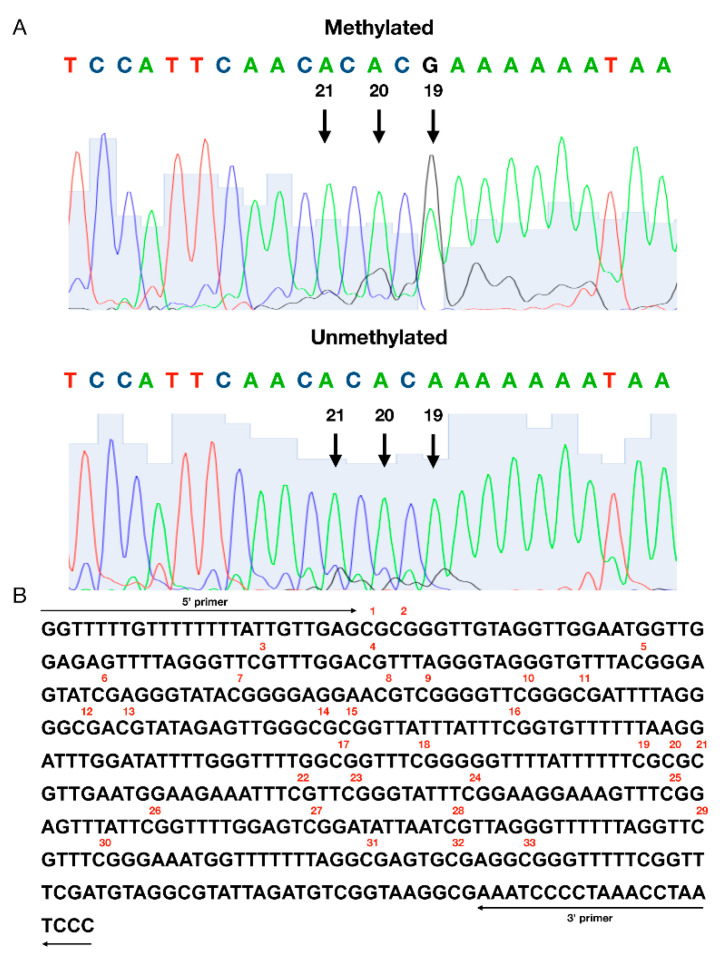
An assessment of a methylation status of individual CpG sites in the *DAT1* promoter. (**A**) The representative result of the positive (top) and negative (bottom) methylation statuses; (**B**) the sequence of the analyzed *DAT1* promoter. Numbers were assigned to individual sites in the studied region starting from 5′. The methylation status of individual CpG sites was detected with a cut-off level at 20% of the G/A + G ratio using 4Peaks software (Mek & Tosj, Amsterdam, The Netherlands).

**Table 1 brainsci-10-00400-t001:** The methylation status of 33 *DAT1* CpG sites in cannabis-dependent and control subjects.

CpG Site	Studied Group	Methylation Status (%)	χ^2^(p)	OR	95% CI (−95%, +95%)	Spearman’s R (p)
**1 ***	**dependent N (201)**	**62%**	**11.689 (0.001)**	**0.528**	**(0.365, 0.763)**	**−0.155 (0.001)**
	**control N (285)**	**46%**				
2	dependent N (201)	78%	1.992 (0.158)	0.739	(0.485, 1.125)	−0.064 (0.159)
	control N (285)	72%				
3	dependent N (201)	86%	4.986 (0.026)	1.921	(1.075, 3.431)	0.101 (0.026)
	control N (285)	92%				
4	dependent N (201)	25%	0.003 (0.952)	1.013	(0.669, 1.533)	0.003 (0.952)
	control N (285)	26%				
5	dependent N (201)	26%	5.357 (0.021)	1.597	(1.072, 2.377)	0.105 (0.021)
	control N (285)	36%				
**6 ***	**dependent N (201)**	**18%**	**16.044 (0.0001)**	**0.309**	**(0.170, 0.562)**	**−0.182 (0.0001)**
	**control N (285)**	**6%**				
7	dependent N (201)	15%	0.289 (0.591)	0.869	(0.522, 1.448)	−0.024 (0.592)
	control N (285)	14%				
8	dependent N (201)	3%	1.499 (0.221)	1.812	(0.691, 4.754)	0.056 (0.222)
	control N (285)	5%				
9	dependent N (201)	36%	0.217 (0.641)	1.093	(0.751, 1.590)	0.021 (0.642)
	control N (285)	38%				
10	dependent N (201)	37%	0.00001 (0.998)	1.000	(0.689, 1.453)	0.0001 (0.998)
	control N (285)	37%				
11	dependent N (201)	3%	2.360 (0.124)	1.979	(0.816, 4.802)	0.069 (0.125)
	control N (285)	7%				
12	dependent N (201)	30%	0.105 (0.746)	1.067	(0.721, 1.580)	0.015 (0.746)
	control N (285)	31%				
13	dependent N (201)	2%	8.037 (0.005)	3.769	(1.417, 10.022)	0.128 (0.005)
	control N (285)	9%				
14	dependent N (201)	85%	0.046 (0.832)	0.947	(0.577, 1.556)	−0.010 (0.831)
	control N (285)	84%				
15	dependent N (201)	83%	0.019 (0.891)	0.967	(0.602, 1.554)	−0.006 (0.891)
	control N (285)	82%				
16	dependent N (201)	67%	2.594 (0.107)	0.733	(0.502, 1.070)	−0.073 (0.108)
	control N (285)	60%				
17	dependent N (201)	28%	0.266 (0.606)	1.110	(0.746, 1.651)	0.023 (0.607)
	control N (285)	31%				
18	dependent N (201)	7%	0.002 (0.969)	0.986	(0.495, 1.964)	−0.002 (0.969)
	control N (285)	7%				
19	dependent N (201)	92%	7.920 (0.005)	3.435	(1.386, 8.511)	0.128 (0.005)
	control N (285)	98%				
20	dependent N (201)	45%	2.558 (0.110)	0.741	(0.513, 1.070)	−0.072 (0.110)
	control N (285)	38%				
21	dependent N (201)	72%	2.441 (0.118)	0.732	(0.495, 1.083)	−0.071 (0.119)
	control N (285)	65%				
22	dependent N (201)	87%	10.045 (0.002)	2.876	(1.461, 5.659)	0.144 (0.001)
	control N (285)	95%				
23	dependent N (201)	19%	0.080 (0.777)	0.935	(0.587, 1.489)	−0.013 (0.777)
	control N (285)	18%				
24	dependent N (201)	70%	0.284 (0.594)	0.899	(0.609, 1.328)	−0.024 (0.595)
	control N (285)	67%				
25	dependent N (201)	25%	5.761 (0.016)	1.632	(1.092, 2.440)	0.109 (0.016)
	control N (285)	35%				
26	dependent N (201)	37%	2.549 (0.110)	1.350	(0.933, 1.953)	0.072 (0.111)
	control N (285)	45%				
27	dependent N (201)	17%	0.143 (0.705)	1.096	(0.681, 1.764)	0.017 (0.706)
	control N (285)	18%				
28	dependent N (201)	73%	6.067 (0.014)	0.611	(0.412, 0.906)	−0.112 (0.014)
	control N (285)	62%				
29	dependent N (201)	25%	1.888 (0.169)	0.738	(0.479, 1.139)	−0.062 (0.170)
	control N (285)	20%				
30	dependent N (201)	10%	0.074 (0.786)	1.084	(0.605, 1.941)	0.012 (0.786)
	control N (285)	11%				
31	dependent N (201)	5%	0.290 (0.590)	1.234	(0.574, 2.653)	0.024 (0.591)
	control N (285)	7%				
32	dependent N (201)	68%	0.004 (0.951)	1.012	(0.686, 1.492)	0.003 (0.951)
	control N (285)	68%				
33	dependent N (201)	73%	2.599 (0.107)	1.413	(0.927, 2.152)	0.073 (0.107)
	control N (285)	79%				

The chi-squared test χ^2^(p); (OR) odds ratio; (CI) Confidence Interval; R(p) Spearman correlation (−95%, +95%). To positions 1 and 6, the Bonferroni correction was applied to get the Bonferroni critical value (*p* = 0.0015).

**Table 2 brainsci-10-00400-t002:** Potential transcription factors capable of binding analyzed CpG sites. The ability of transcription factors to bind individual regions was assessed with different similarity rates, i.e., 100% (**a**), 95% (**b**), or 85% (**c**).

Matrix Similarity Rate	CpG Position	Transcription Factor
(**a**) 100%	3	PAX5
	33	PAX5
(**b**) 95%	1	GCF
	3	PAX5
	11	RXR-alpha
	19	c-Ets-2
	20	c-Ets-2
	22	PAX5, p53, Sp1
	25	AP-2alphaA
	28	NFI/CTF
	33	PAX5
(**c**) 85%	1	GCF, E2F1
	3	PAX5, p53
	5	AhR
	6	GR alpha
	11	TFII-I, STAT4, RXR-alpha
	13	PAX5, p53
	19	GR alpha, c-Ets-2, E2F1, GCF
	20	c-Ets-2, E2F1, GCF
	22	PAX5, p53, E2F1, Sp1
	25	GR alpha, AP-2alphaA, NF-AT2
	26	PAX5, p53
	28	ENKTF1, EBF, E2F1, NFI/CTF
	33	PAX5, p53, E2F1

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
