# Peer review of "Contribution of Dopamine Transporter Gene Methylation Status to Cannabis Dependency"

_brainsci, 2020, doi:10.3390/brainsci10060400_

Round 1

Reviewer 1 Report

The authors aimed to investigate the association between methylation of the promoter of the dopamine transporter gene, DAT1, and cannabis use disorder. This is an important area of research. As they mention, the legalization of cannabis has led to growing use across the world, yet much remains unknown about the underlying biology of cannabis use disorders, and epigenetics are likely to be an important component of that biology. They performed bisulfite conversion and methylation-specific PCR to examine the methylation status in cases with cannabis use disorder and controls in peripheral blood. In addition, they assessed transcription factor binding using the TRANSFAC database. Some strengths include the transcription factor analysis, however, for this and many other parts of the paper, some important details are missing. Suggestions are included below. 

Suggestions:

Language:

  • In general, substantial language editing is needed throughout the manuscript. Many sentences were not grammatically correct and there are several typos (e.g. missing parenthesis). In addition, addiction and addict are not preferred terminology when discussing substance use disorders and they should be corrected throughout the manuscript. It is also unclear what diagnostic criteria were used, so the terminology should also be accurate in terms of diagnostic criteria.

Abstract: 

  • When discussing transcription factors, it would be more accurate to say they "have been shown to play a role in the development of the nervous system" rather than they are "responsible for the development of the nervous system" 

Introduction:

  • When discussing DNA methylation, you should mention we understand it functions to repress/activate genes but it may also have other functions since it is not fully understood
  • Try to be more accurate with wording throughout given how much is unknown about the biological implications of epigenetics. For example, instead of "DNA methylation may lead to some behavioral abnormalities" "DNA methylation has been associated with behavioral abnormalities" is more accurate based on the reviews cited.
  • Instead of Marijuana (abundant in THC) say: Marijuana, which has high levels of THC. As is it sounds like marijuana is an abundant type of THC.
  • Other studies to cite related to alcohol dependence:

Dopamine Transporter Gene Methylation is Associated with Nucleus Accumbens Activation During Reward Processing in Healthy but not Alcohol-Dependent Individuals. Muench C, Wiers CE, Cortes CR, Momenan R, Lohoff FW. Alcohol Clin Exp Res. 2018 Jan;42(1):21-31. doi: 10.1111/acer.13526. Epub 2017 Nov 27.PMID: 29030974 

  • It would be useful to know how many CpG sites could have been analyzed (why did you pick the 33 you picked, are those all of them?)
  • "This study takes up one of those challenges" seems inaccurate since it is cross-sectional and thus does not determine if methylation is a cause or consequence of cannabis use or cannabis use disorder

Methods:

  • The paper does not define the diagnostic criteria used to determine case status. Also, how was it determined whether they had abstained for 3 months prior?
  • Figure quality needs improvement.
  • There were no corrections or considerations for important covariates such as smoking, BMI, other substance use disorders and other psychiatric disorders.
  • It is unclear what a similarity rate is, does this involve DNA methylation or genetic variants?

Results:

  • Addict is incorrect terminology in Table 1.

Discussion:

  • It's worth writing more in depth about methylation findings for DAT1 in other substance use disorders (in addition to opioids) as its association with many substance use disorders suggest broader implications for this subset of psychiatric disorder. It is also important given the high rates of comorbidity among psychiatric disorders, and the potential for comorbidities as confounding factors. Some studies on cocaine, smoking and alcohol that should be cited are included below:

Association analysis between polymorphisms in the dopamine D2 receptor (DRD2) and dopamine transporter (DAT1) genes with cocaine dependence. Lohoff FW, Bloch PJ, Hodge R, Nall AH, Ferraro TN, Kampman KM, Dackis CA, O'Brien CP, Pettinati HM, Oslin DW.Neurosci Lett. 2010 Apr 5;473(2):87-91. doi: 10.1016/j.neulet.2010.02.021. Epub 2010 Feb 17.PMID: 20170711

Neural correlates of attentional bias for smoking cues: modulation by variance in the dopamine transporter gene. Wetherill RR, Jagannathan K, Lohoff FW, Ehrman R, O'Brien CP, Childress AR, Franklin TR.Addict Biol. 2014 Mar;19(2):294-304. doi: 10.1111/j.1369-1600.2012.00507.x. Epub 2012 Oct 12.PMID: 23061530 

  • When discussing why studies should include females and other races, it should be noted that genetic and epigenetic markers to vary by sex and race and that the conclusions from this study are thus limited.
  • A discussion of the replicability of candidate gene methylation studies should also be added. These findings should be followed up, as some candidate gene methylation studies have been replicable while others have not.

Author Response

Dear Reviewer,

We would like to thank you for your valuable comments on the article. Below you will find our reply to your review. All changes are with a description or a comment and changes have been made to the manuscript (track changes in the tracking group on the review tab).

The authors aimed to investigate the association between methylation of the promoter of the dopamine transporter gene, DAT1, and cannabis use disorder. This is an important area of research. As they mention, the legalization of cannabis has led to growing use across the world, yet much remains unknown about the underlying biology of cannabis use disorders, and epigenetics are likely to be an important component of that biology. They performed bisulfite conversion and methylation-specific PCR to examine the methylation status in cases with cannabis use disorder and controls in peripheral blood. In addition, they assessed transcription factor binding using the TRANSFAC database. Some strengths include the transcription factor analysis, however, for this and many other parts of the paper, some important details are missing. Suggestions are included below.

Suggestions:

Language:

In general, substantial language editing is needed throughout the manuscript. Many sentences were not grammatically correct and there are several typos (e.g. missing parenthesis). In addition, addiction and addict are not preferred terminology when discussing substance use disorders and they should be corrected throughout the manuscript. It is also unclear what diagnostic criteria were used, so the terminology should also be accurate in terms of diagnostic criteria.

Thank you for these suggestions. The manuscript has been proofread by the translator.

“Addiction” has been changed to “dependency or substance use disorder” as it fitted best throughout the manuscript.

Diagnostic criteria have been described in more detail – page 3, line 114-117.

Abstract:

When discussing transcription factors, it would be more accurate to say they "have been shown to play a role in the development of the nervous system" rather than they are "responsible for the development of the nervous system"

Thank you for this suggestion – this sentence has been rephrased – page 1, line 37-38.

Introduction:

When discussing DNA methylation, you should mention we understand it functions to repress/activate genes but it may also have other functions since it is not fully understood

Thank you for this suggestion – the sentence has been added – page 2, line 52-53.

Try to be more accurate with wording throughout given how much is unknown about the biological implications of epigenetics. For example, instead of "DNA methylation may lead to some behavioral abnormalities" "DNA methylation has been associated with behavioral abnormalities" is more accurate based on the reviews cited.

Thank you for this suggestion – the sentence has been rephrased – page 1, line 46.

Instead of Marijuana (abundant in THC) say: Marijuana, which has high levels of THC. As is it sounds like marijuana is an abundant type of THC.

Thank you for this suggestion – the sentence has been rephrased – page 2, line 78.

Other studies to cite related to alcohol dependence:

Dopamine Transporter Gene Methylation is Associated with Nucleus Accumbens Activation During Reward Processing in Healthy but not Alcohol-Dependent Individuals. Muench C, Wiers CE, Cortes CR, Momenan R, Lohoff FW. Alcohol Clin Exp Res. 2018 Jan;42(1):21-31. doi: 10.1111/acer.13526. Epub 2017 Nov 27.PMID: 29030974

Thank you for this suggestion – the citation has been added – number 26 on reference list

It would be useful to know how many CpG sites could have been analyzed (why did you pick the 33 you picked, are those all of them?)

Thank you for this question. The analysed 447-bp fragment of the DAT1 promoter is placed in the position between -610 and -1057 from the transcription starting site. We agree that there are many other CpG sites which could have be methylated, however, we believe that the analysed fragment is sufficient and representative enough to prove our hypothesis. Furthermore, because one of limitations of this method of measuring methylation is obtaining the best possible primers, in the case of the DAT1 promoter it could be designed only for the analysed fragment.

"This study takes up one of those challenges" seems inaccurate since it is cross-sectional and thus does not determine if methylation is a cause or consequence of cannabis use or cannabis use disorder

Thank you for this suggestion – this sentence has been removed.

Methods:

The paper does not define the diagnostic criteria used to determine case status. Also, how was it determined whether they had abstained for 3 months prior?

Thank you for this question. The diagnostic criteria have been added in the materials and method section. The patients were recruited from a closed addiction treatment centre with  round the clock care and their abstinence was determined based on the amount of time they were in therapy.

Figure quality needs improvement.

Figure quality was improved.

There were no corrections or considerations for important covariates such as smoking, BMI, other substance use disorders and other psychiatric disorders.

Thank you for this interesting feedback. The analysed patients went through a diagnostic procedure and substance dependency was their only diagnosis. As for possible other substance use these data are being currently analysed and will be published in the future. The same is the case for other covariates.

It is unclear what a similarity rate is, does this involve DNA methylation or genetic variants?

The description of prediction of transcription factor binding affinity was clarified in both the text and the material and methods section – page 4, line 152-160

Results:

Addict is incorrect terminology in Table 1.

Thank you for this suggestion. The terminology has been changed in Table 1.

Discussion:

It's worth writing more in depth about methylation findings for DAT1 in other substance use disorders (in addition to opioids) as its association with many substance use disorders suggest broader implications for this subset of psychiatric disorder. It is also important given the high rates of comorbidity among psychiatric disorders, and the potential for comorbidities as confounding factors. Some studies on cocaine, smoking and alcohol that should be cited are included below:

Association analysis between polymorphisms in the dopamine D2 receptor (DRD2) and dopamine transporter (DAT1) genes with cocaine dependence. Lohoff FW, Bloch PJ, Hodge R, Nall AH, Ferraro TN, Kampman KM, Dackis CA, O'Brien CP, Pettinati HM, Oslin DW.Neurosci Lett. 2010 Apr 5;473(2):87-91. doi: 10.1016/j.neulet.2010.02.021. Epub 2010 Feb 17.PMID: 20170711

Neural correlates of attentional bias for smoking cues: modulation by variance in the dopamine transporter gene. Wetherill RR, Jagannathan K, Lohoff FW, Ehrman R, O'Brien CP, Childress AR, Franklin TR.Addict Biol. 2014 Mar;19(2):294-304. doi: 10.1111/j.1369-1600.2012.00507.x. Epub 2012 Oct 12.PMID: 23061530

Thank you for this interesting suggestion. The information you required has been added – page 9, line 258-261, reference 46 and 47.

When discussing why studies should include females and other races, it should be noted that genetic and epigenetic markers to vary by sex and race and that the conclusions from this study are thus limited.

Thank you for this suggestion. The sentence has been added – page 9, line 251-252.

A discussion of the replicability of candidate gene methylation studies should also be added. These findings should be followed up, as some candidate gene methylation studies have been replicable while others have not.

Thank you for this suggestion. “There is very little research concerning epigenetic and in particular methylation changes in addiction research. To our best knowledge there is no study analysing methylation of few genes in the same group of subjects, incorporating full phenotypic description of addiction (i. e. substance(s), age of onset, number of relapse, and many others), psychometric testing and single nucleotide polymorphism (SNPs) data so the question of replicability is still open”. Added in the text, page: 9, line : 263-267.

Reviewer 2 Report

The study presented in the manuscript "Contribution of Dopamine Transporter Gene Methylation Status to Cannabis Addiction" by Grzywach et al., looks at the DAT1 promoter methylation status in peripheral blood leucocytes of patients compared to healthy individuals. Although they dont find an overall change in the methylation status, they do see changes in some sites. 

The study is a simple study , though new but very preliminary. 

Some comments are as follows: 

1) line # 173, "we analyzed the Tf ability to bind individual regions with different similarity rates, i.e, 100, 95, 85%". What exactly do the authors mean by this? 

2)Line #164, the authors write "However, there was no rule that would indicate higher or lower methylation of particular sites". And then the authors describe the difference between the sites. What exactly do the authors mean by this? It seems contradictory to their own staetments.

3) Why have the authors not checked the expression of DAT1 gene? This experiment can add more information and strength to the manuscript.

 4)Is it possible to add methylation tracts figure ?  for the sites where the difference is seen? 

Author Response

Dear Reviewer,

We would like to thank you for your valuable comments on the article. Below you will find our reply to your review. All changes are with a description or a comment and changes have been made to the manuscript (track changes in the tracking group on the review tab).

Comments and Suggestions for Authors

The study presented in the manuscript "Contribution of Dopamine Transporter Gene Methylation Status to Cannabis Addiction" by Grzywach et al., looks at the DAT1 promoter methylation status in peripheral blood leucocytes of patients compared to healthy individuals. Although they dont find an overall change in the methylation status, they do see changes in some sites. 

The study is a simple study, though new but very preliminary. 

Some comments are as follows: 

  • line # 173, "we analyzed the Tf ability to bind individual regions with different similarity rates, i.e, 100, 95, 85%". What exactly do the authors mean by this? 

The clarity of the statement was improved in the text. Moreover, additional clarification was conducted by extending the description in the material and methods section – page 4, line 152-160

2)Line #164, the authors write "However, there was no rule that would indicate higher or lower methylation of particular sites". And then the authors describe the difference between the sites. What exactly do the authors mean by this? It seems contradictory to their own staetments.

Thank you for this comment. We agree that the above-mentioned statement could be misleading. It has been removed from the text.

3) Why have the authors not checked the expression of DAT1 gene? This experiment can add more information and strength to the manuscript.

Thank you for this comment. The expression experiment was not performed at that time. This type of analysis is planned for the future ant then the data will be analysed collectively.

 4) Is it possible to add methylation tracts figure ?  for the sites where the difference is seen? 

Thank you for this comment. We improved the quality of the figure.